# Determinants of physical activity participation among U.S. adolescents aged 12–17: A study of key factors

RuChen She[1], Dongmei Wang[2*]

1 Wanjiang University of Technology, Ma'anshan, Anhui, China, 2 Anhui University of Technology, Ma'anshan, Anhui, China

* 395846158@qq.com

## Abstract

This study examines the influence of six key factors—family structure, poverty level, parental education, mental health, family resilience, and childhood adversity—on the participation of U.S. adolescents (ages 12–17) in moderate-to-vigorous physical activity (MVPA) for at least 60 minutes per day. Using data from the 2018–2023 health surveys across all 50 states and the District of Columbia, the study applies significance difference analysis, random forest regression, and cluster analysis to assess the impact and interaction effects of these factors on adolescent physical activity. The results indicate that mental health, family resilience, and poverty level have the most significant influence on adolescents' physical activity participation, with strong interactions among the variables. Cluster analysis further reveals disparities in physical activity participation across states, with certain states showing significant imbalances due to factors such as poverty and family support. These findings underscore the importance for policymakers to consider the unique socioeconomic contexts and policy environments of different states when promoting youth physical activity, advocating for tailored approaches to enhance adolescent physical activity levels.

## Introduction

It is widely recognized that higher levels of physical activity play a crucial role in promoting the physical and mental health development of adolescents [1]. In 2020, the World Health Organization (WHO) recommended that adolescents should accumulate at least 60 minutes of moderate-to-vigorous physical activity (MVPA) daily [2]. Regular physical activity not only benefits the health of adolescents but also positively impacts academic performance. For example, Ryan D. Burns found in his study that adolescents who participated in one or more sports teams reported higher academic achievements compared to those who did not engage in sports teams [3]. In terms of mental health, more frequent physical activity and sports participation significantly improve adolescents' well-being while reducing symptoms of anxiety and depression

**Data availability statement:** The datasets generated and/or analysed during the current study are available in the [NSCH] repository, [https://www.childhealthdata.org/].

**Funding:** This work is funded by Anhui University of Technology through grant number WTS202427 awarded to DW.

**Competing interests:** The authors declare that they have no competing interests.

[4]. Furthermore, Allana G. LeBlanc and colleagues, through their research on seven health indicators (high cholesterol, hypertension, metabolic syndrome, obesity, low bone density, depression, and injuries), suggested that children and adolescents aged 5–17 should engage in at least 60 minutes of moderate-intensity physical activity daily and increase the intensity of activity when possible, including exercises that strengthen muscles and bones [5]. Therefore, appropriate physical activity has a significant positive impact on various aspects of adolescents' academic performance, mental health, and physical health.

From a global perspective, adolescent participation in physical activity exhibits inequalities across gender, region, and socio-economic backgrounds [6]. These disparities are primarily influenced by factors such as economic affordability, time constraints, accessibility to resources and facilities, and family background [7]. These barriers directly impact adolescent physical activity (PA) behaviors and may become key environmental variables influencing youth activity patterns, including child care, school recess, physical education, mental health, healthcare environment, family income, and family dynamics [8]. For example, Russell R. Pate and colleagues, in their study of determinants of adolescent physical activity participation, found that during key transition periods from elementary to middle school and from middle to high school, adolescents' physical activity is influenced by a complex interaction of psychosocial, family, school, community, and physical environmental factors [9]. Additionally, Noriko Motoki and colleagues, through their analysis of the Sagawa Sports Foundation 2019 "Sport-Life" survey, further highlighted the importance of family and social backgrounds in adolescent sports participation. The study collected data via written questionnaires on gender, age, grade, household income, family composition, lifestyle habits, and participation in organized sports and MVPA. It found that family support, especially from parents, significantly influenced adolescents' participation in physical activities [10]. Overall, social background and family environment collectively impact adolescent physical activity levels, and these factors should be considered in interventions aimed at promoting youth participation in physical activity.

This study, based on data from 2018–2023 on the proportion of adolescents in each state engaging in over 60 minutes of daily physical activity, utilizes cluster analysis and radar chart analysis to explore the impact of six factors—family structure, poverty level, parental education, mental health, family resilience, and childhood adversity—on adolescent physical activity participation. Through these analyses, we identify similarities and differences in physical activity participation among different states and gain a deeper understanding of the complex socio-economic and geographical factors behind these disparities. The findings provide valuable insights for guiding policy development and intervention strategies aimed at promoting adolescent physical activity participation.

## Data and methodology

### Data sources

The data for this study were obtained from the U.S. Child Health Survey, funded and guided by the Maternal and Child Health Bureau (MCHB) of the Health Resources

and Services Administration (HRSA). We selected data from five survey cycles: 2018–2023, which cover all 50 states and the District of Columbia, with a focus on adolescents aged 12–17.The survey question used to assess physical activity was: "In the past week, how many days did this child engage in exercise, sports, or physical activities for at least 60 minutes?" In accordance with the 2020 World Health Organization (WHO) guidelines, which recommend that adolescents should accumulate at least 60 minutes of moderate-to-vigorous physical activity (MVPA) daily, this study selected the percentage of adolescents aged 12–17 who met this physical activity threshold as the core data metric for analysis.

Additionally, the data were grouped based on several family-related variables, including Family Structure (Both Parents Married, Single Parent), Poverty Level (Poverty 0–99%, Poverty 100–199%, Poverty 200–399%, Poverty 400 and Above), Parental Education (High school or GED, Some College or Technical School, College Degree or Higher), Mental Health (Mental Health CSHCN Yes, Mental Health CSHCN No), Family Resilience (Family Resilience Yes, Family Resilience No), and Childhood Adversity (No Adverse Childhood Events, One Adverse Childhood Event, Two or More Adverse Childhood Events).

To ensure data integrity, random forest imputation was used to address missing values. Subsequently, sensitivity analysis was conducted through scatter plots to evaluate the fit and stability of the imputed data compared to the original dataset. Specifically, the original data were compared with the imputed data, and scatter plots were created for each variable. Each scatter plot visualized the relationship between the original data (X-axis) and the imputed data (Y-axis). By analyzing these scatter plots, we could visually assess the correlation between the imputed and original data. Ideally, the imputed data should exhibit a strong linear relationship with the original data, meaning the data points in the scatter plot should be closely clustered around the diagonal line (y = x). This would indicate that the imputation process successfully preserved the structure and distribution of the original data. Moreover, the scatter plots allowed us to identify potential outliers or imputation errors. If the scatter plots revealed significant deviations between the imputed and original data, or if the data points were widely scattered, this could suggest that the imputation method introduced substantial error or inaccurate imputed values.

In this study, the scatter plots for all variables showed a strong linear relationship between the imputed data and the original data, with most imputed data points tightly clustered around the diagonal line. This indicates that the imputation method was effective and did not introduce significant systematic errors. Consequently, we conclude that the imputation method did not result in notable deviations from the data structure, validating the reliability of the imputed results (Fig 1).

## Research methodology

In this study, we first employed different statistical tests—independent sample t-tests, Analysis of Variance (ANOVA), and Kruskal-Wallis tests—to systematically assess the statistical significance of various factors influencing adolescent participation in 60 minutes or more of daily moderate-to-vigorous physical activity (MVPA) across five research cycles.We used the t-test to analyze parent variables with only two subgroups, such as Mental Health, Family Resilience, and Family Structure. For parent variables with three or more subgroups, such as Poverty Level, Parental Education, and Childhood Adversity, we applied ANOVA and Kruskal-Wallis tests. ANOVA is appropriate for normally distributed data and effectively evaluates mean differences between groups, whereas the Kruskal-Wallis test is used for non-normally distributed data, overcoming the limitations imposed by assumptions about data distribution.Since ANOVA and Kruskal-Wallis tests can only identify overall differences between groups but do not specify which groups differ significantly, we further conducted Tukey's HSD post-hoc test. Tukey's test controls for Type I errors while enabling pairwise comparisons of means, which improves the accuracy and reliability of the analysis. By combining ANOVA with post-hoc testing, we provided a more refined analytical framework to identify the key factors affecting MVPA participation rates.

Next, we used Random Forest Regression to model the multidimensional data and quantify the relative importance of each parent variable on MVPA participation rates. To enhance the model's generalizability and stability, we performed Grid Search Optimization in combination with Cross-Validation for hyperparameter tuning. After extensively searching the

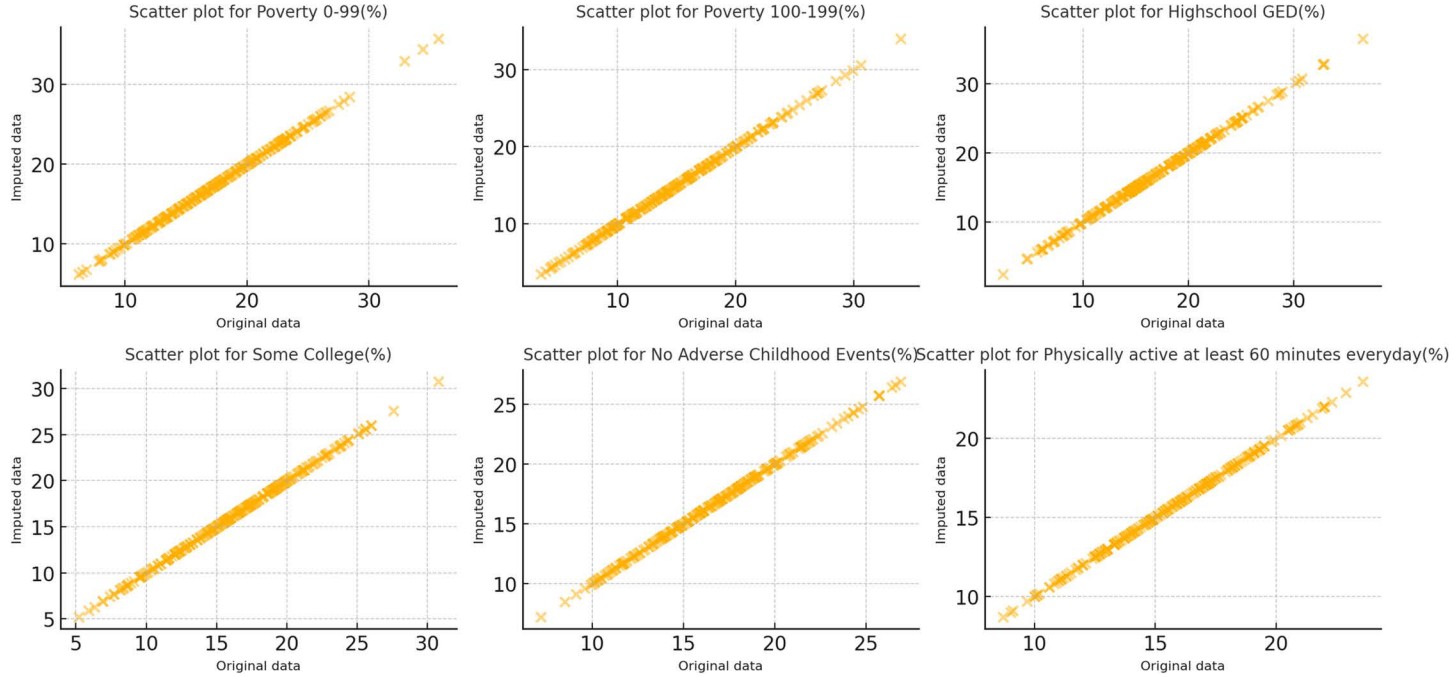

**Fig 1. Scatter plot of imputed vs original data. Sensitivity analysis of data imputation using random forest.** A strong linear relationship between original and imputed data suggests structural integrity was maintained.

parameter space, we determined the optimal combination of parameters, including the number of decision trees (n_estimators = 377), minimum samples required to split an internal node (min_samples_split = 2), minimum samples required to be at a leaf node (min_samples_leaf = 4), maximum number of features (max_features = log2), and maximum depth of the trees (max_depth = 10). This optimization process ensured the robustness of the model while effectively preventing overfitting.

The model evaluation results indicate that the final model achieved an average absolute error (MAE) of 1.58 on the test set, with a prediction accuracy of 88.56%. Furthermore, the actual maximum depth of the model was 8, which is lower than the predefined maximum depth limit (10), further confirming the model's robustness. By plotting a scatter comparison between the predicted and actual values, we observed that most of the predicted values aligned closely with the actual values. This suggests that the model provides a good fit and is capable of accurately capturing the key factors influencing MVPA participation rates. These findings demonstrate that the model offers a scientifically reliable approach to data modeling for this study (Fig 2).

In analyzing regional differences, hierarchical clustering was applied to the standardized data. Hierarchical clustering determines the similarity between states by calculating the Euclidean distance across multiple features. By progressively merging similar states and representing these relationships with a dendrogram, we were able to derive groupings of states. The Ward method was used during the clustering process, which minimizes the variance within each cluster, making it suitable for measuring features of different scales.Specifically, the hierarchical clustering was performed using the linkage function from the SciPy library, based on the standardized data. To select the optimal clustering solution, we evaluated different max_d values, ranging from 2 to one less than the total sample size. The optimal clustering configuration was chosen based on the Silhouette Score, with the best result occurring at a max_d value of 10, yielding a silhouette score of 0.699.

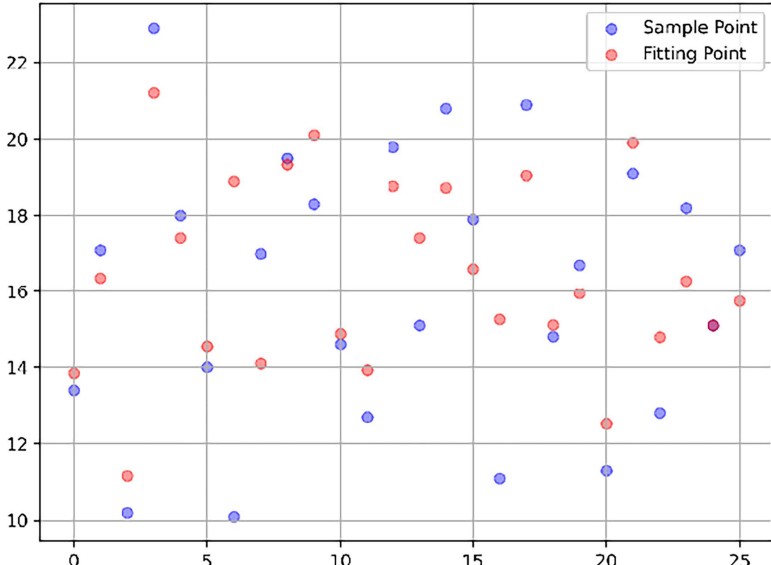

**Fig 2. Original vs predicted data.** Model validation of the random forest algorithm using a scatter plot of predicted vs. observed values. The close alignment of most points along the diagonal suggests strong model fit.

## Results

### Significance analysis of variables

Among the six selected variables—Family Structure, Poverty Level, Parental Education, Mental Health, Family Resilience, and Childhood Adversity—we conducted significance tests for the subgroups of each variable over a five-year period. For the data involving multiple comparisons, post-hoc tests were performed. The results are as follows:Tables 1 and 2.

The results of the study indicate that mental health, family resilience, poverty level, and childhood adversity have a significant impact on MVPA participation rates, while the effects of parental education and family structure show annual variations. However, significance testing mainly compares whether there are statistically significant differences in the means or distributions of the target variable (MVPA) across different categorical variables, but it does not allow us to directly infer which specific factor plays a decisive role. Therefore, it is necessary to incorporate Random Forest Regression, which measures the contribution of each variable to prediction error (e.g., Mean Squared Error (MSE)), to better assess the relative importance of each factor.

### Variable feature importance analysis

Through the Random Forest model, we performed an importance ranking of the six selected variables in this study. The results show that the degree of influence each variable has on the target variable (MVPA participation rate) varies significantly:

Family Structure emerged as the most influential variable, accounting for approximately 35% of the total importance, indicating its strongest impact on predicting physical activity participation. Poverty level ranked second with an importance of 22%, suggesting that economic status has a significant effect on the target variable. Parental education level and mental health both exhibited similar importance, ranging from 12% to 14%, indicating that they moderately influence an individual's physical activity participation behavior. Family resilience and childhood adversity showed relatively lower importance, with values of approximately 5% and 4%, respectively, demonstrating their smaller independent contributions within the model.

**Table 1. Significance_test_results.**

|  | Family Structure | Poverty Level | Parental Education | Mental Health | Family Resilience | Childhood Adversity |
|---|---|---|---|---|---|---|
| 2018–2019 | 0.108 | 0.016 | 0.072 | 0.001 | 0.000 | 0.235 |
| 2019–2020 | 0.871 | 0.003 | 0.012 | 0.000 | 0.000 | 0.260 |
| 2020–2021 | 0.251 | 0.010 | 0.007 | 0.000 | 0.000 | 0.009 |
| 2021–2022 | 0.100 | 0.001 | 0.026 | 0.000 | 0.000 | 0.000 |
| 2022–2023 | 0.142 | 0.037 | 0.160 | 0.000 | 0.000 | 0.005 |

Significance testing of multiple variables over a five-year period. Values represent p-values used to assess whether each variable has a statistically significant effect on the target variable.

**Table 2. Post-hoc Tukey HSD test.**

|  | Poverty Level | | | | | | Parental Education | | | Childhood Adversity | | |
|---|---|---|---|---|---|---|---|---|---|---|---|---|
| 2018–2019 | 0.163 | 0.064 | 0.176 | 0.976 | 1.000 | 0.970 | 0.063 | 0.174 | 0.883 | 0.226 | 0.750 | 0.620 |
| 2019–2020 | 0.015 | 0.003 | 0.094 | 0.947 | 0.902 | 0.606 | 0.012 | 0.118 | 0.629 | 0.228 | 0.701 | 0.675 |
| 2020–2021 | 0.003 | 0.012 | 0.154 | 0.975 | 0.488 | 0.749 | 0.008 | 0.262 | 0.318 | 0.834 | 0.043 | 0.156 |
| 2021–2022 | 0.002 | 0.094 | 0.297 | 0.533 | 0.220 | 0.939 | 0.080 | 0.068 | 0.997 | 0.004 | 0.000 | 0.201 |
| 2022–2023 | 0.229 | 0.003 | 0.014 | 0.362 | 0.663 | 0.961 | 0.123 | 0.312 | 0.868 | 0.024 | 0.009 | 0.930 |

Post hoc analysis using Tukey's HSD test for datasets involving multiple comparisons. The results identify significant pairwise differences after correction for multiple testing.

However, feature importance analysis primarily measures the contribution of individual variables and does not directly reveal the complex interaction effects between variables. Therefore, certain variables—such as mental health and family resilience—although showing significant differences in the significance analysis, may exhibit relatively low independent importance in the feature importance analysis. This could be due to the fact that their interaction effects play a more substantial role within the overall model.Next, we will conduct an interaction effects analysis between variables to verify this hypothesis.

In the influence mechanism of socio-economic and family environment factors, there are often complex interaction effects between variables. That is, the effect of one variable on the target variable may depend on the value of another variable. To further explore these interactions, we used contour plots to visualize how different combinations of variables affect the target variable (Fig 3). This method provides a clear visual representation of how the response value of the target variable changes when two variables interact.In this process, we created combinations of the six variables in pairs, resulting in 15 unique combinations (Fig 4):

The significance of variable interactions was mainly judged by the changes in the color gradient and the magnitude of changes in the contour values. A noticeable change in color from dark to light indicates a large difference in the target variable under that variable combination, suggesting a strong interaction effect. The more drastic the numerical change, the more significant the joint impact of that variable combination on the target variable. Based on these criteria, we selected the following four variable combinations with significant interaction effects for further analysis: Poverty level and Family Structure、Mental Health and Family Structure、Poverty level and Parental Education、Childhood Adversity and Family Resilience.

(1) Low poverty level + stable family structure (e.g., two-parent family) → highest physical activity participation (lightest color); high poverty level + unstable family structure (e.g., single-parent family) → lowest physical activity participation (darkest color). At low poverty levels, the influence of family structure is minimal, meaning that whether the family is two-parent or single-parent, individuals' physical activity participation remains at a high level. At high poverty levels,

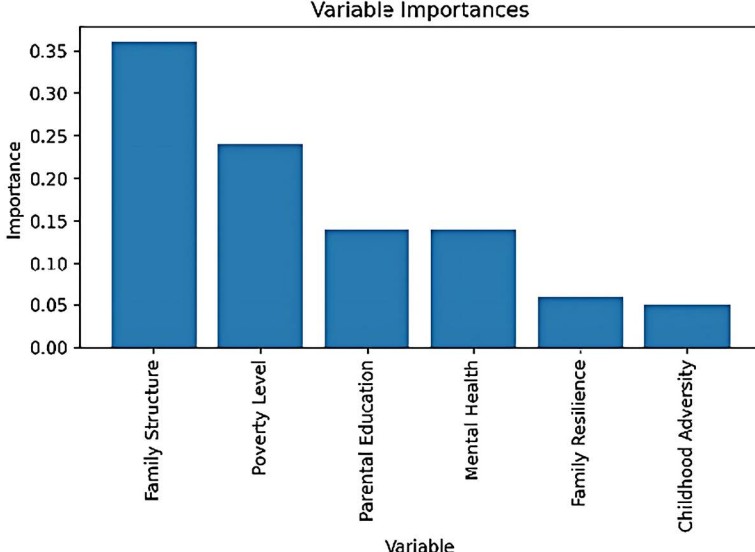

**Fig 3. Variable importance. Variable importance ranking based on the random forest model.** The results show significant differences in the contribution of each variable to the prediction of the target variable.

the impact of family structure dramatically increases, with individuals from single-parent families showing a significant drop in physical activity participation. This suggests that the combined effect of economic poverty and family instability amplifies the impact on health behaviors.

(2) Good mental health + stable family structure → higher physical activity participation; poor mental health + unstable family structure → lowest physical activity participation (darkest color). When mental health is good, the influence of family structure is relatively small, meaning that individuals with good mental health maintain a high level of physical activity participation regardless of their family structure. However, when mental health is poor, the impact of family structure becomes significant. Specifically, in single-parent families, individuals with poor mental health show a significant drop in physical activity participation, while in two-parent families, their participation level improves.

(3) High parental education level + low poverty level → highest physical activity participation (lightest color); low parental education level + high poverty level → lowest physical activity participation (darkest color). When poverty levels are low, the influence of parental education level is minimal, meaning that individuals maintain a relatively high level of physical activity participation. However, when poverty levels are high, the influence of parental education level becomes significant. Individuals with higher parental education levels are able to maintain a certain level of physical activity participation, while those with lower parental education levels experience a sharp decline in participation.

(4) Low childhood adversity (fewer adverse childhood experiences) + high family resilience → highest physical activity participation (lightest color); high childhood adversity + low family resilience → lowest physical activity participation (darkest color). When childhood adversity is low, the influence of family resilience is minimal, meaning that individuals with fewer adverse childhood experiences maintain a high level of physical activity participation regardless of their family resilience. However, when childhood adversity is high, the influence of family resilience becomes significant. Among individuals with high childhood adversity, those with stronger family resilience have higher physical activity participation than those with lower family resilience.

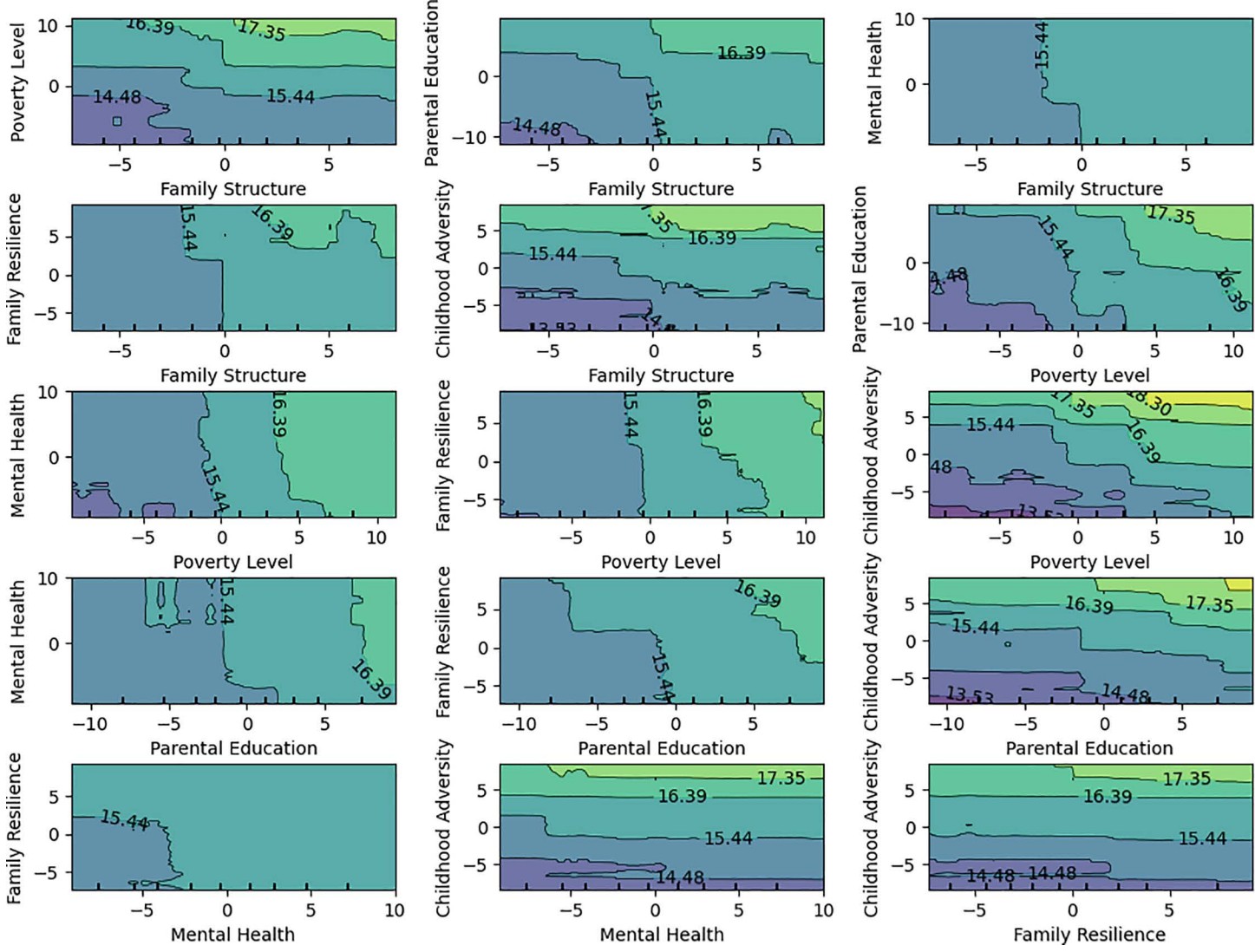

**Fig 4. Interaction analysis.** Contour plots visualizing the joint effects of variable pairs on the target variable. All 15 combinations of the six selected variables were analyzed to explore interaction patterns and response surface changes.

Based on the above analysis, the initial hypothesis of this study was confirmed: the variables do not independently affect the youth MVPA level, but rather exhibit strong interaction effects and synergistic influences. Moreover, some variables with lower importance (such as mental health and family resilience) have an indirect impact on the target variable, primarily through the mediating effects of other variables like poverty level, family structure, and childhood adversity, which show higher importance in the random forest analysis. Additionally, the effects of mental health and family resilience may depend on specific interaction effects, which is why they were not assigned higher weights in the independent variable importance ranking of the random forest model.

## Cluster analysis

This section discusses the results of the cluster analysis and radar charts, aiming to analyze the clustering results of youth physical activity data across states. The goal is to assess how various factors (such as family structure, poverty level,

parental education, mental health, family resilience, and childhood adversity) influence the proportion of adolescents aged 12–17 engaging in more than 60 minutes of physical activity per day. Based on this, we aim to identify state groups with similar characteristics and explore how different variables affect youth physical activity levels. The cluster results are as follows (Fig 5):

The dendrogram displays two main clusters. The cluster on the left (orange area) and the cluster on the right (green area) show clear separation in Euclidean distance, indicating significant differences in MVPA participation and related socioeconomic factors. Based on the clustering results from the dendrogram, we divide all states into two main clusters:

Cluster 1 includes states from different regions of the East, South, West, and Midwest, such as Maryland (East), Colorado (West), Texas (South), South Carolina (South), Nevada (West), and others. Geographically, these states are spread across various regions of the United States, showing a broad geographic distribution. Most of these states are located in areas with relatively stable or strong economic development, such as California, Colorado, New Jersey, and Maryland [11].

Cluster 2 also includes states from different geographic regions, such as Florida (Southeast), Illinois (Midwest), California (West), Texas (South), and others. These states have a similarly broad geographic distribution, but compared to Cluster 1, this cluster may be more concentrated in the Southeast and Midwest regions. These states typically face lower economic levels, such as higher poverty rates, lower median household incomes, and poorer educational and health resources. Further analysis will be conducted using radar charts, combined with relevant policies, to explore these factors in detail [12] (Fig 6).

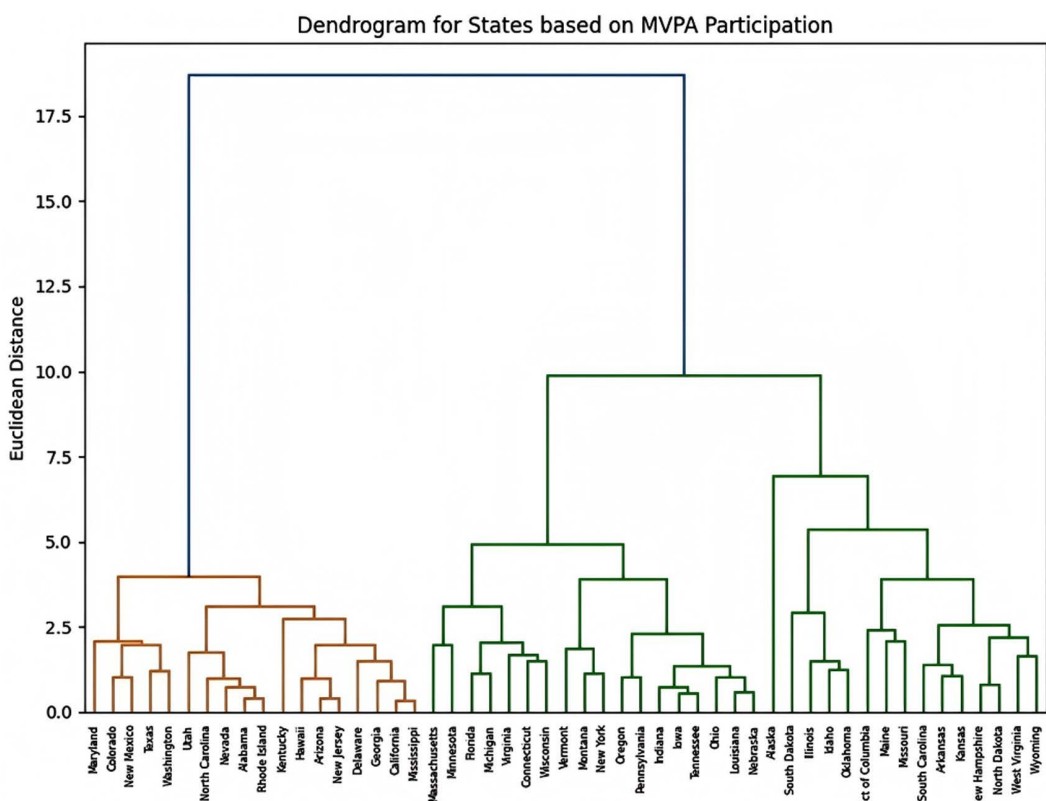

**Fig 5. Dendrogram for states on MVPA participation.** Dendrogram based on hierarchical clustering, illustrating groups of states with similar characteristics and how different variables influence the target variable.

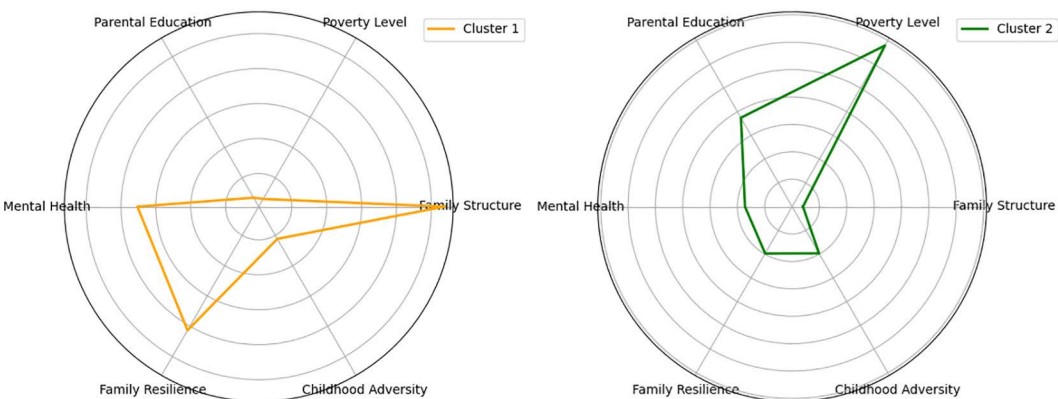

**Fig 6. Radar chart for clusters based on MVPA participation.** Radar charts of states within the same cluster, used to assess the relative influence of different variables on the target variable within each group.

According to the radar chart analysis, the most significant factor influencing daily MVPA (moderate to vigorous physical activity) for adolescents in Cluster 1 states is family structure, followed by mental health and family resilience. The impact of poverty level, parental education, and childhood adversity is relatively small. Research shows that these less influential factors are closely related to government support policies. For example, in 2019, California passed the California Family Strengthening Act, which aims to provide social support, mental health resources, family education programs, and economic assistance to strengthen families' ability to cope with stress and challenges. The act enhanced family resilience through community collaboration projects, family education, and mental health interventions, while promoting adolescent health and well-being. In the same year, California also launched the California Adverse Childhood Experiences (ACEs) Awareness Initiative, designed to train healthcare providers and community workers to identify and address adverse childhood experiences, providing early intervention and treatment for affected groups to reduce the long-term impact of these adversities on adolescent physical and mental health. Additionally, Colorado's Strengthening Families Initiative focuses on building resilience in families, especially low-income families, helping them improve their ability to cope with stress. The initiative provides mental health support, parenting education, and community services aimed at better supporting children's healthy development. In New Jersey, the Parent Education and Family Literacy Program, implemented in 2018, enhances parental education levels, particularly in parenting and community support, for low-income and impoverished communities. This program provides additional resources for parents, enabling them to better support the educational and health development of their adolescents. These policies and programs offer more opportunities for adolescents from low-income and high-risk groups to participate in physical activities, helping them overcome economic, social, and health barriers, and thus contributing to improving their overall health.

In Cluster 2, the factors that most significantly influence daily MVPA (moderate to vigorous physical activity) for adolescents are poverty level and parental education, followed by mental health, family resilience, and childhood adversity, while family structure has a relatively small impact on adolescents. These states share common economic characteristics, such as higher poverty rates, lower median household incomes, a shortage of educational resources, and significant socioeconomic disparities. These economic factors often limit opportunities for adolescents to engage in physical activities, thereby affecting their health levels.To address these issues, states in Cluster 2 have implemented a series of policies and programs aimed at improving adolescent health and well-being. For example, Mississippi launched the Mississippi Strengthening Families Program in 2016, which helps low-income families strengthen their resilience to adversity through

family support services. The program includes mental health interventions, parenting education, and economic assistance, with the goal of enhancing the resilience of adolescent families when facing challenges. In 2019, Alabama introduced the Alabama Strengthening Families Initiative, which offers mental health services, parenting training, and community support to help families improve their ability to adapt to stress and challenges, thus improving the environment in which adolescents grow.Meanwhile, Arkansas's Arkansas Behavioral Health Integration Program integrates mental health services into the primary healthcare system and provides psychological support to adolescents through health insurance. The program also collaborates with schools and communities to ensure that adolescents, especially those from low-income families, have access to timely mental health services. In addition, Arkansas has implemented the Arkansas Childhood Trauma Response Program, which aims to reduce the impact of adverse childhood experiences and trauma by providing necessary mental health support to families and helping them cope with psychological trauma.In 2020, Louisiana launched the Louisiana Family Assistance Program, which provides financial assistance to low-income families, helping them access more social services, including healthcare, education, and sports facilities. By increasing family income, the program aims to offer adolescents more opportunities to engage in physical activities, thereby promoting their physical and mental health.

These measures and initiatives reflect the states' efforts to create a more supportive health and development environment for adolescents from low-income and high-risk groups through multi-faceted interventions. By helping them overcome economic and social barriers, these programs aim to enhance their levels of physical activity and overall health.

In summary, states in Cluster 1 generally possess a strong economic foundation and educational resources, which enable them to implement comprehensive policies and programs aimed at enhancing the physical and mental health of adolescents and increasing their participation in physical activities. These policies, through family support, mental health interventions, parental education, economic aid, and the reduction of adverse childhood experiences, increase opportunities for adolescents to engage in moderate-to-vigorous physical activity (MVPA), thereby improving their overall health.

In contrast, states in Cluster 2 face economic challenges, such as higher poverty rates, limited educational resources, and significant socio-economic disparities. However, these states also implement policy interventions and support programs to assist adolescents, particularly those from low-income and high-risk backgrounds, in overcoming socio-economic barriers and providing more opportunities for physical activity. These policies promote adolescent health development and physical activity participation by improving family economic status, offering mental health support, enhancing parental education, and mitigating the effects of adverse childhood experiences.

These policies and programs indeed highlight the critical role of multi-level support in promoting the physical and mental health of adolescents, particularly in low-income and impoverished communities. Through the joint efforts of governments and communities, not only can family economic conditions be improved, but mental health support, educational resources, and increased opportunities for physical activity participation can also be provided. However, despite the positive interventions already in place, there remains room for improvement and enhancement, which will be addressed in the subsequent discussion.

## Discussion

This study aims to explore the determinants influencing the daily participation of U.S. adolescents aged 12–17 in 60 minutes of moderate to vigorous physical activity (MVPA), as well as the interactions among these factors. The results indicate that mental health, family resilience, and poverty level are significant factors affecting adolescent physical activity participation. Additionally, strong interactions were found between poverty level and family structure, mental health and family structure, poverty level and parental education, as well as childhood adversity and family resilience.

Adolescents' attitudes toward physical activity (PA) also influence the maintenance and improvement of MVPA [13], and therefore, the study of adolescents' PA can largely reflect the impact of related factors on MVPA. Parental education level only shows a significant impact under specific circumstances, and there is no consensus in the literature regarding

the relationship between parental education level and adolescent PA. For example, Sherar et al. found that girls with less educated mothers had lower levels of physical activity [14], while Vázquez-Nava [15] and Ball K [16] found the opposite—higher maternal education was associated with lower levels of adolescent physical activity. However, this study found a strong interaction between parental education level and family poverty level. Specifically, when poverty level is high, parental education level significantly influences adolescent MVPA: individuals with higher-educated parents tend to maintain higher levels of physical activity, while those with lower-educated parents show a sharp decline in participation. In contrast, when poverty level is low, the effect of parental education level is less pronounced. Therefore, based on the analysis of this study, parental education level indirectly affects adolescents' MVPA through family poverty level. As society progresses, when family poverty levels gradually decrease, parental education level may become a more decisive factor influencing adolescents' daily MVPA. This could be because, with the increasing global emphasis on adolescent physical activity, parents with higher education levels are more likely to provide scientifically grounded guidance and advanced ideas for the physical and mental health development of their children [16–18].

.In terms of family poverty levels, related research has confirmed that parents' economic status and social standing significantly impact adolescents' PA [19]. As adolescents grow older, participating in physical activities becomes more complex and economically costly (e.g., sports club fees), which may reduce the likelihood of adolescents from low-income families engaging in PA [20]. Research by Kellstedt et al [21]. also supports this finding, showing that family income has a significant effect on adolescent sports participation; the higher the income, the higher the rate of adolescent participation in sports, nearly four times greater. Additionally, FKS Mathisen et al [22].conducted a 27-year longitudinal study on the determinants of adolescents' vigorous physical activity (VPA) during leisure time. This study reported that family income is a decisive factor in influencing adolescents' participation in high-intensity physical activities (VPA).

Regarding mental health, Marlier M et al [23]. argued that there is a significant positive correlation between adolescents' PA and mental health. However, it is important to note that this view is based on the premise that adolescents' PA typically represents a selected leisure activity aimed at entertainment and enjoyment, which helps improve mood and self-perception, rather than being a compulsory activity, such as household chores, gardening, or work activities [24]. The expressions of mental health and childhood adversity share certain similarities, which may be due to the fact that childhood adversity is a risk factor for subsequent mental health issues, and there is a certain degree of correlation between the two [25].

There is limited research on the impact of family resilience on adolescents' MVPA, but studies on family resilience have become a focus of research over the past decade [26]. Across the United States, efforts are being made to improve the specific methods and resources needed to enhance family resilience [27]. Evidence-based programs and policies in healthcare, education, and human services aimed at strengthening family resilience and connections may promote the thriving of American children [28]. In this study, family resilience showed a significant positive correlation with adolescents' MVPA. Strong family support and the resilience demonstrated by families in the face of adversity greatly facilitate adolescents' participation in physical activities. Furthermore, when adolescents experience high levels of childhood adversity, family resilience plays a key role. Good family resilience can mitigate the negative effects of childhood adversity, allowing adolescents to maintain daily MVPA even under high childhood adversity.

In the cluster analysis section, through hierarchical clustering and radar chart analysis, we were able to clearly identify the similarities and differences in sports participation patterns among states. A detailed analysis of various aspects such as the economy, education, culture, and demographics of each state revealed that sports participation is influenced by multiple factors. Combining these factors, we can conclude that the differences in adolescent sports participation and influencing factors across states mainly stem from their unique socio-economic backgrounds and policy support [11,29]. For example, Mississippi is one of the states with the highest poverty rate in the U.S., with the adolescent poverty rate higher than the national average. Adolescents from low-income families face numerous barriers in terms of access to sports facilities, nutrition, and healthcare, limiting their opportunities to engage in MVPA [30]. Alabama also has a high

poverty rate, particularly in rural areas where the lack of sports facilities and health resources restricts opportunities for youth to participate in physical activities [31]. Arkansas has a median family income lower than the national average, and many families face financial instability. Due to lower incomes, many adolescents do not have the opportunity to participate in extracurricular sports activities or health programs, limiting their physical activity participation [32]. Louisiana has a low median family income, particularly in the low-income communities of cities such as New Orleans. Due to economic pressures, adolescents lack the necessary support to engage in MVPA, which affects their health and quality of life [32]. Although West Virginia has some industrial resources, the state's poverty-stricken areas have poor educational resources and lack modern school sports facilities and health education programs, resulting in fewer opportunities for youth to engage in physical activities [31,33]. Therefore, when policymakers develop programs to promote adolescent sports participation, they should fully consider the actual context of each state and adopt differentiated, context-specific measures to enhance the overall physical activity levels of the adolescent population [33,34].

## Recommendations

### Increase adolescent physical activity participation in low-income families

Although the poverty level is relatively low in Cluster 1 states, some low-income families still face barriers to engaging in physical activity. Gaps in parental education may also impact adolescents' opportunities to participate in sports activities. At the government and community levels, more family support programs should be offered, such as health education courses and parenting education, to help low-income families improve their health literacy and provide more opportunities for their children to engage in physical activities. Additionally, greater collaboration between schools and communities can increase the availability of extracurricular activities and sports facilities, especially in low-income areas, ensuring that all adolescents have access to healthy activities.

### Strengthen support for low-income families

Governments and communities in most of the states in Cluster 2 should strengthen their support for low-income families by providing more sports facilities and activities for adolescents. For example, offering free sports facilities and supporting community sports programs, particularly in impoverished areas, will help improve the physical health of adolescents. Additionally, increasing mental health resources in schools and communities to provide counseling and interventions can reduce the impact of mental health issues on adolescents' participation in physical activities.

### Provide social support for adolescents from unstable family structures

For adolescents from unstable family structures, schools and communities should enhance family support services, such as family counseling, parent-child relationship training, and family intervention programs. These services will help family members learn how to better cope with stress, communicate effectively, and establish a stable family environment.

### Enhance family resilience

For families with low resilience, community support networks should be strengthened through community centers, social service agencies, and volunteer programs, providing social and emotional support to low-income families. Parenting guidance and family communication skills should be offered, teaching parents how to create a positive family atmosphere, thereby improving children's mental health and social adaptability.

## Conclusion

This study analyzed the key factors influencing adolescents aged 12–17 in their daily participation in 60 minutes of moderate to vigorous physical activity (MVPA), as well as the interactions between these variables, using differential

analysis and random forest regression models. The results indicate that mental health, family resilience, and poverty level have the most significant impact on youth physical activity participation, and there are strong interaction effects among the variables. Cluster analysis further revealed differences in physical activity participation across states, with some states showing significant imbalances in participation due to factors such as poverty and family support. The findings emphasize that policymakers, when promoting youth physical activity, should consider the unique socio-economic contexts and policy support of each state, and adopt differentiated measures to improve adolescents' physical activity levels.

## Strengths and limitations

This study utilized health survey data from all 50 states and Washington, D.C. in the United States from 2018 to 2023. The broad coverage of the data allows for a comprehensive reflection of adolescent physical activity participation across different socio-economic backgrounds. Furthermore, by combining multiple methods, such as significance difference analysis, random forest regression, and cluster analysis, the study not only enhances the reliability and interpretability of the results but also uncovers key factors influencing youth physical activity participation and regional differences.

However, this study has some limitations. First, although the data spans multiple years, it is based on survey responses, which may introduce inaccuracies or response biases, especially when dealing with subjective factors such as mental health and family resilience. This could affect the precision of the results. Second, the study mainly focuses on quantitative analysis and lacks qualitative research on the social and cultural context behind adolescent physical activity participation, as well as the personal subjective experiences. This somewhat limits a deeper understanding of the mechanisms through which these factors influence participation.

Future research could further reduce biases caused by subjective responses by conducting more detailed longitudinal surveys and in-depth interviews. Additionally, exploring the interactive effects between different variables will help to more comprehensively reveal the complex mechanisms influencing adolescent physical activity participation. Moreover, incorporating qualitative research to better understand how different social backgrounds and cultures impact adolescent physical activity will provide policymakers with more targeted recommendations.

## Author contributions

**Conceptualization:** Ruchen She.

**Data curation:** Ruchen She.

**Formal analysis:** Ruchen She.

**Funding acquisition:** Ruchen She.

**Investigation:** Ruchen She.

**Methodology:** Ruchen She.

**Project administration:** Ruchen She, Dongmei Wang.

**Resources:** Ruchen She.

**Software:** Ruchen She.

**Supervision:** Ruchen She.

**Validation:** Ruchen She.

**Visualization:** Ruchen She.

**Writing – original draft:** Ruchen She.

**Writing – review & editing:** Ruchen She.

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
