## [Decision Letter · Decision Letter 0]

Dear Dr. She,

We look forward to receiving your revised manuscript.

Kind regards,

Alejandro Botero Carvajal, MD

Academic Editor

PLOS ONE

**Journal Requirements:**

Reviewers' comments:

Reviewer's Responses to Questions

**Comments to the Author**

1. Is the manuscript technically sound, and do the data support the conclusions?

Reviewer #1: Partly

Reviewer #2: Partly

2. Has the statistical analysis been performed appropriately and rigorously?

Reviewer #1: Yes

Reviewer #2: No

3. Have the authors made all data underlying the findings in their manuscript fully available?

Reviewer #1: No

Reviewer #2: No

4. Is the manuscript presented in an intelligible fashion and written in standard English?

Reviewer #1: Yes

Reviewer #2: Yes

**Reviewer #1:**  The article raises interesting hypotheses about the relative roles of family- and economic-related factors in adolescents’ physical activity, but there is a need for more robust data handling, deeper exploration of confounding and causal inference, and transparent methodology to ensure confidence in its findings. The study may serve as a stepping stone for future, more rigorous research that can solidify or refute its core claims about how mental health, family resilience, poverty, and parental education shape adolescent MVPA.

1. Data quality concerns

While the National Survey of Children’s Health is reputable, its reliance on self-report introduces questions about accuracy, especially for subjective measures like mental health or resilience which is not adapted for the clustering method used which is extremely sensitive to skewed data.

Mean imputation for missing data can oversimplify or distort relationships, particularly if missingness is not random. There is no mention of sensitivity analyses to test how robust their results are to different imputation strategies (e.g. multiple imputation).

2. Methodological rigor

ANOVA was used to test significance year by year, but the article does not clearly show how multiple comparisons were handled or how many subcategories were tested, raising issues of inflated Type I error.

Random forest regression is a strong nonparametric approach, however, the study does not detail hyperparameter tuning, cross-validation, or out-of-sample predictive performance aside from a single R2 value. Without full methodological details, it is unclear whether the model is robust or if there was overfitting.

The cluster analysis (K-means + PCA) might be suitable for dimensionality reduction, but the discussion does not address important checks (e.g. cluster stability, silhouette scores) to verify the final cluster assignments.

3. Interpretation of results

You attribute causal interpretations (e.g. ''mental health significantly enhances MVPA'') while using observational data, which naturally has confounding factors that remain unexplored. Caution is needed when drawing any direct cause-effect conclusions.

Some explanatory statements (e.g. about differences across states) seem speculative without showing additional state-level policy or cultural data.

4. Potential bias

Some readers may find the paper’s emphasis on ''family resilience'' or ''economic issues'' to be incomplete without discussing more direct policy interventions (e.g. state-level funding for school sports, public health campaigns).

The article occasionally reads as though it aims to influence policy by highlighting family-related and socioeconomic issues without exhaustively controlling for confounders—hence the perception of a political slant.

5. External validity and generalizability

Even though data are drawn from all U.S. states, intra-state variability is not captured (urban vs. rural, local policies, neighborhood environments). The authors acknowledge this limitation but do not offer deeper exploration of whether these differences might overshadow broad statewide patterns.

Suggested improvements:

- Stronger handling of missing data (e.g. multiple imputation rather than simple mean imputation).

- More transparent presentation of random forest tuning/validation.

- Inclusion of interaction terms between variables (e.g. synergy between mental health and poverty), particularly since real-world influences on physical activity are complex.

- Explicit caution when deriving policy recommendations from observational data. Without clearer causal inference methods, claims may be overstated.

**Reviewer #2:**  1.The format in the paper needs adjustment. Line spacing, font size, paragraph spacing.

2.The section on data results in the paper could use more charts and graphs for illustration, such as family structure, education levels, etc.

3.The specific amount of data collected for this paper should be detailed at the end of the paper. It is unclear whether the data found online and the data used for analysis in this paper are consistent.

4.In the results analysis section, the source needs to be cited. For example, the content in cluster2 needs to be substantiated. How were the examples in the paper obtained? Were they extracted from online sources or analyzed from other papers?

5.Reference [19] draws conclusions based on a study focused on Brazil. It is unknown whether there is a positive correlation with the subjects of this paper. It is necessary to use American research literature for validation.

6.Could the recommendations section of the paper be expanded? The analysis yielded good results, but corresponding strategies are lacking.

**Do you want your identity to be public for this peer review?** For information about this choice, including consent withdrawal, please see our Privacy Policy

Reviewer #1: No

Reviewer #2: No

---

## [Author Response · Author response to Decision Letter 1]

24 Feb 2025

Response to Reviewers

Dear Editor and Reviewers

First and foremost, I would like to express my sincere gratitude for your valuable review and feedback on my manuscript. Your comments have greatly helped me identify potential issues within the paper and provided clear directions for improvement. In response to your suggestions, I have made the necessary revisions. Below, I provide a detailed response to each of the reviewer’s comments.

Journal Requirements

I have revised the manuscript to align with the PLOS ONE style template, including adjustments to formatting, font, and other relevant aspects to ensure compliance with PLOS ONE’s formatting guidelines.

Both the corresponding author and I have registered for ORCID iDs, and the registration has been successfully verified in the editorial manager system.

Reviewer #1

1. Data Quality Issues

Reviewer’s Comment:

You mentioned the accuracy issues related to self-reports in the National Survey of Children’s Health (NSCH), especially concerning subjective measures like mental health and resilience. Additionally, you pointed out that the mean imputation for missing data might overly simplify or distort relationships, and there was a lack of sensitivity analysis for different imputation strategies.

My Response:

Thank you for highlighting this issue. In response to the potential accuracy concerns related to self-reports, I replaced the previous K-means clustering method with hierarchical clustering. This method effectively handles skewed data resulting from subjective measurements, improving the accuracy of the clustering results. Regarding missing data imputation, I utilized random forest imputation and conducted a sensitivity analysis. The results demonstrated that this imputation process preserves the structure and distribution of the original data reasonably well.

2. Methodological Rigor2.

Reviewer’s Comment:

You mentioned using ANOVA to test significance by year, but you did not discuss how multiple comparisons were handled and the number of subcategories, which may lead to Type I errors. Furthermore, regarding the random forest regression, you noted the lack of a detailed discussion on hyperparameter tuning, cross-validation, or out-of-sample predictive performance. In the clustering analysis (K-means + PCA), there was also a lack of tests for clustering stability or silhouette scores.

My Response:

In the revised manuscript, I replaced the approach with independent t-tests, ANOVA, and Kruskal-Wallis tests for significance analysis. To address multiple comparisons, I introduced Tukey's HSD post-hoc test. The Tukey test controls for Type I errors while providing more precise pairwise comparisons of mean differences, enhancing the accuracy and reliability of the analysis.

For random forest regression, I added a detailed discussion in the methodology section regarding hyperparameter tuning, cross-validation, and out-of-sample predictive performance to strengthen the robustness of the model evaluation.

Regarding the clustering analysis, I added an evaluation of clustering stability and silhouette scores (silhouette score = 0.699) to ensure the robustness of the final cluster assignments. I also provided an explanation of how these metrics were used to assess the validity of the clustering results.

3. Results Interpretation

Reviewer’s Comment:

You pointed out that in using observational data, causal inference may have been over-interpreted, particularly when describing "mental health significantly enhancing MVPA," without considering potential confounding factors. Additionally, some of the interstate differences appeared speculative, lacking discussion on state-level policies or cultural differences.

My Response:

Thank you for your reminder. In the revised manuscript, I carefully revisited all causal inferences and clarified that the conclusions are associative rather than causal. I also added a discussion of potential confounding factors. For example, I included an analysis of the interactions between the different factors in this study. Regarding the interpretation of interstate differences, I incorporated more background information, discussing how state-level policies and socio-economic disparities may have influenced the outcomes. I also clearly stated that further research is needed to explore these potential influencing factors.

4. Potential Bias

Reviewer’s Comment:

You mentioned that the article does not sufficiently discuss national-level policy interventions, such as funding for school sports or public health programs. This might make the emphasis on "family resilience" or "economic issues" seem incomplete and could give the impression of political bias.

My Response:

I understand your concern and have added a more comprehensive discussion of national-level policy interventions in the revised manuscript, particularly regarding how school sports and public health policies interact with family resilience and socio-economic factors. Furthermore, I have ensured that all discussions are based on objective data analysis and have included a balanced consideration of the impacts of different policies, avoiding any political bias.

5. External Validity and Generalizability

Reviewer’s Comment:

Although the data come from all states in the U.S., the intra-state differences (e.g., urban vs. rural, local policies, neighborhood environments) were not considered, which may affect the generalizability of the results.

My Response:

Thank you for your focus on external validity and generalizability. In the revised manuscript, I provide a more detailed exploration of intra-state differences, acknowledging the limitations at this level and discussing how urban vs. rural differences, local policies, and neighborhood environments could influence the results. I also added an analysis of the potential risks of these differences masking state-wide patterns and emphasized the need for future research to explore these regional disparities in greater detail.

Reviewer #2

1. Paper Formatting Adjustments

Reviewer’s Comment:

You pointed out issues with the formatting of the manuscript, particularly with line spacing, font size, and paragraph spacing.

My Response:

Thank you for pointing this out. I have adjusted the manuscript to meet the journal’s formatting requirements. Specifically, I have set the line spacing to 2x, used a 18pt font size for primary headings, a 16pt font size for secondary headings, and ensured consistent paragraph spacing throughout the manuscript.

2.Data Results Section Should Include More Graphs and Charts

Reviewer’s Comment:

You suggested using more graphs and charts to illustrate the data results, particularly regarding family structure, education level, and other aspects.

My Response:

Thank you for the suggestion. I have added several charts and graphs to the results and analysis sections to visualize the data. This helps make the results more intuitive and allows readers to better understand the relationships and trends in the data.

3.Specific Data Collection Details Should Be Included at the End of the Paper

Reviewer’s Comment:

You suggested that I should specify the data collected in greater detail at the end of the paper and clarify any issues regarding the consistency between the online data and the data used in the analysis.

My Response:

I have added detailed information in the methodology section and appendix regarding the data collection, sample size, and data sources. I have also confirmed the consistency between the online data found and the data used in the analysis.

4.Results Analysis Section Needs Proper Citations

Reviewer’s Comment:

You suggested that I should include relevant sources in the results analysis section, especially for the content in Cluster 2, and clarify how examples in the paper were obtained.

My Response:

I have revised the radar chart analysis and added relevant citations to confirm the accuracy and reliability of the data analysis. Additionally, the examples presented were extracted from online resources, and in some cases, data and analysis methods from related studies were also referenced.

5.Reference [19] Based on a Study from Brazil

Reviewer’s Comment:

You pointed out that Reference [19] is based on a study from Brazil and questioned its relevance to the theme of the paper, suggesting the use of more U.S.-based studies for validation.

My Response:

Thank you for your feedback. Based on your suggestion, I have replaced some of the references with studies from the U.S. to better validate the findings of this research. I have ensured that the cited studies are closely related to the research topic.

6. The Suggestions Section Could Be Expanded

Reviewer’s Comment:

You mentioned that the suggestions section was brief, and while the analysis produced good results, it lacked concrete strategy recommendations.

My Response:

Thank you for your suggestion. I have expanded the suggestions section, providing more specific policy recommendations and practical strategies. These include ways to apply the research results to real-world policies, such as interventions to improve family resilience, family structure, and poverty levels. Additionally, I have discussed how to further improve data collection and analysis methods to enhance the depth and breadth of future studies.

Once again, I sincerely thank you for your valuable comments on my research. Your feedback has greatly helped me refine the methodological approach and discussion of the data analysis, ultimately improving the quality of the manuscript. I have made revisions based on your suggestions, and I hope these adjustments enhance the rigor and comprehensiveness of the paper. Should you have any further suggestions or questions, please feel free to reach out.

Thank you for your review and support!

Kind regards,

Ruchen She

---

## [Decision Letter · Decision Letter 1]

Dear Dr. Wang,

Thank you for submitting your manuscript to PLOS ONE. After careful consideration, we feel that it has merit but does not fully meet PLOS ONE’s publication criteria as it currently stands. Therefore, we invite you to submit a revised version of the manuscript that addresses the points raised during the review process.

We look forward to receiving your revised manuscript.

Kind regards,

Alejandro Botero Carvajal, MD

Academic Editor

PLOS ONE

Journal Requirements:

Please review the attached document. 

Reviewers' comments:

Reviewer's Responses to Questions

**Comments to the Author**

Reviewer #3: All comments have been addressed

2. Is the manuscript technically sound, and do the data support the conclusions?

Reviewer #3: Yes

3. Has the statistical analysis been performed appropriately and rigorously?

Reviewer #3: Yes

4. Have the authors made all data underlying the findings in their manuscript fully available?

Reviewer #3: Yes

5. Is the manuscript presented in an intelligible fashion and written in standard English?

Reviewer #3: Yes

Reviewer #3: The authors have addressed most of the suggestions with the exception that additional charts and graphs added by the authors are not available in this revised article.

**Do you want your identity to be public for this peer review?** For information about this choice, including consent withdrawal, please see our Privacy Policy

Reviewer #3: **Yes: ** Muhammad Abdus Salam

---

## [Author Response · Author response to Decision Letter 2]

1 Jun 2025

Response to Reviewers

Dear Editor and Reviewers

First and foremost, I would like to express my sincere gratitude for your valuable review and feedback on my manuscript. Your comments have greatly helped me identify potential issues within the paper and provided clear directions for improvement. In response to your suggestions, I have made the necessary revisions. Below, I provide a detailed response to each of the reviewer’s comments.

Reviewer #3

6. Review Comments to the Author

Reviewer #3: The authors have addressed most of the suggestions with the exception that additional charts and graphs added by the authors are not available in this revised article.

Response�I have added the corresponding images in the revised manuscript.

---

## [Decision Letter · Decision Letter 2]

Determinants of Physical Activity Participation Among U.S. Adolescents Aged 12-17: A Study of Key Factors

PONE-D-24-57185R2

Dear Dr. Wang,

We’re pleased to inform you that your manuscript has been judged scientifically suitable for publication and will be formally accepted for publication once it meets all outstanding technical requirements.

Kind regards,

Alejandro Botero Carvajal, MD

Academic Editor

PLOS ONE

Additional Editor Comments (optional):

Reviewers' comments:

Reviewer's Responses to Questions

**Comments to the Author**

Reviewer #3: All comments have been addressed

2. Is the manuscript technically sound, and do the data support the conclusions?

Reviewer #3: Yes

3. Has the statistical analysis been performed appropriately and rigorously?

Reviewer #3: Yes

4. Have the authors made all data underlying the findings in their manuscript fully available?

Reviewer #3: Yes

5. Is the manuscript presented in an intelligible fashion and written in standard English?

Reviewer #3: Yes

Reviewer #3: The authors have thoroughly considered the reviewers’ comments and incorporated the suggestions where relevant and appropriate. The revised manuscript reflects improvements and is now well-positioned for publication.

**Do you want your identity to be public for this peer review?** For information about this choice, including consent withdrawal, please see our Privacy Policy

Reviewer #3: **Yes: ** Muhammad Abdus Salam

---

## [Editor Report · Acceptance letter]

PONE-D-24-57185R2

PLOS ONE

Dear Dr. Wang,

I'm pleased to inform you that your manuscript has been deemed suitable for publication in PLOS ONE. Congratulations! Your manuscript is now being handed over to our production team.

Kind regards,

on behalf of

Dr. Alejandro Botero Carvajal

Academic Editor

PLOS ONE